# UNSUPERVISED ORDERING FOR MAXIMUM CLIQUE

## ABSTRACT

We propose an unsupervised approach for learning vertex orderings for the maximum clique problem by framing it within a permutation-based framework. We transform the combinatorial constraints into geometric relationships such that the ordering of vertices aligns with the clique structures. By integrating this clique-oriented ordering into branch-and-bound search, we improve search efficiency and reduce the number of computational steps. Our results demonstrate how unsupervised learning of vertex ordering can enhance search efficiency across diverse graph instances. We further study the generalization across different sizes.

## 1 INTRODUCTION

Unsupervised Learning (UL) is an emerging paradigm for solving Combinatorial Optimization (CO) problems. While Supervised Learning (SL) requires expensive labelled data, and Reinforcement Learning (RL) struggles with sparse rewards and high training variance, leading to unstable performance, UL offers a promising alternative Min et al. (2023).

The Maximum Clique Problem (MCP) is of fundamental importance in graph theory and combinatorial optimization, with significant theoretical and practical implications. Formally, given an undirected graph $G(V, E)$, where $V$ is the set of vertices and $E$ is the set of edges, the MCP seeks the largest subset $C \subseteq V$ such that $\forall u, v \in C, \{u, v\} \in E$. In other words, the induced subgraph $G[C]$ is a complete graph, and the goal is to maximize $|C|$, the cardinality of the clique. MCP is not only NP-hard but also hard to approximate, since no $O(n^{1-\varepsilon})$-approximation is possible unless P = NP Engebretsen and Holmerin (2000); Khot (2001); Zuckerman (2006). The MCP has wide-ranging applications, including social network analysis, where it helps uncover tightly connected communities, and bioinformatics, where it is used to identify dense clusters in protein interaction networks Bomze et al. (1999).

Exact algorithms for the MCP primarily follow the branch-and-bound framework. Among these, a common strategy is to color the vertices in a specific order for computing upper bounds and guiding vertex selection Tomita and Seki (2003); Tomita et al. (2010); San Segundo et al. (2011); Konc and Janezic (2007). Other methods include iterative deepening with sub-clique information, or use MaxSAT-based bounding Wu and Hao (2015). However, these algorithms mainly rely on hand-crafted features to design effective pruning rules and branching strategies. Recently, there has been a paradigm shift towards data-driven approaches, where machine learning techniques are employed to build efficient search strategies. Among these data-driven methods, UL shows particular promise because it can leverage the inherent structural patterns in graphs without requiring expensive labelled training data.

Several approaches have tackled the MCP using UL by framing it as a binary classification task Karalias and Loukas (2020). Recent advances have focused on two key areas: developing more sophisticated graph neural network architectures and designing novel loss functions Karalias et al. (2022). These approaches aim to learn a function $f_\theta : G(V, E) \to [0, 1]^n$ that maps an input graph to vertex-level probabilities, optimizing the model to identify vertices that belong to the maximum clique.

Here, we propose an alternative approach that learns vertex ordering rather than binary assignments for MCP. Graph vertex ordering is a foundational concept in combinatorial optimization on graphs, including the MCP, where an appropriate vertex permutation can significantly influence the efficiency of exact search algorithms. Traditional approaches often rely on hand-crafted heuristics, such as degree-based ordering Carraghan and Pardalos (1990); Tomita et al. (2010); San Segundo et al.

(2011); Jiang et al. (2017); Szabó and Zavalnij (2018). Recently, graph reordering has also attracted interest in the machine learning community, particularly for enhancing the efficiency of graph neural networks Arai et al. (2016); Balaji and Lucia (2018); Merkel et al. (2024). These studies highlight that suitable vertex permutations can substantially impact algorithmic performance—for example, by exposing dense substructures or minimizing irregular memory access patterns.

Consider the graph and its matrix representations shown in Figure 1. Our goal is to identify potential cliques by reordering vertices. Given a graph with $n$ nodes and adjacency matrix $A \in \mathbb{R}^{n \times n}$, the matrix $\mathbf{M}(A) = J - I - A$ represents non-adjacent vertex pairs with 1s and adjacent pairs with 0s, where $J \in \mathbb{R}^{n \times n}$ is the all-ones matrix and $I \in \mathbb{R}^{n \times n}$ is the identity matrix, $\mathbf{M}(A)$ is the adjacency matrix of the complement graph.

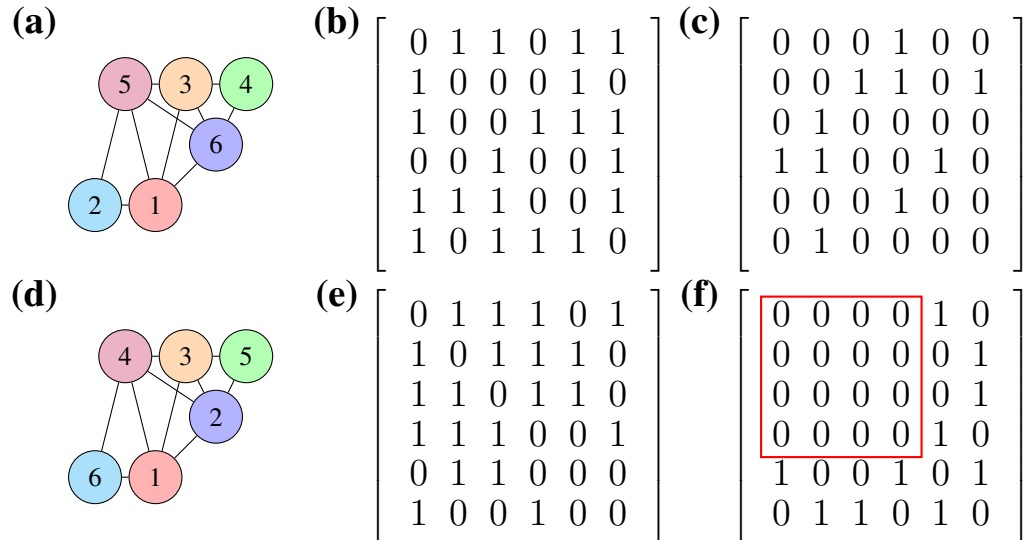

Figure 1: Graph representations and their corresponding matrices. (a) The original graph, (b) the corresponding adjacency matrix $A$, (c) $\mathbf{M}(A) = J - I - A$ (where $J$ is the all-ones matrix and $I$ is the identity matrix); (d) graph (a) with reordered nodes, (e) the corresponding adjacency matrix $A'$, (f) $\mathbf{M}(A') = J - I - A'$.

Now, consider the two different vertex orderings illustrated in Figure 1 (a) and (d). Their corresponding adjacency matrices, denoted as $A$ and $A'$, are shown in Figure 1 (b) and (e). The matrices $\mathbf{M}(A)$ and $\mathbf{M}(A')$ are presented in Figure 1 (c) and (f), respectively. In $\mathbf{M}(A')$, adjacent vertices (represented by 0s) are successfully clustered in the upper-left corner, as highlighted by the red box. This clustering effectively reveals potential clique members, since vertices within a clique must be adjacent to each other, corresponding to the concentrated region of 0s in $\mathbf{M}(A')$. Thus, if we can find an optimal vertex ordering that places the clique nodes at the front, the clique structures will be revealed by the concentrated pattern of 0s in the transformed matrix $\mathbf{M}(A')$.

The reordering can be formally expressed as $\mathbf{M}(A') = \mathbf{P}^T \mathbf{M}(A) \mathbf{P}$, where $\mathbf{P} \in \mathbb{R}^{n \times n}$ is a permutation matrix. This formulation allows us to optimize the ordering of the vertices directly through a permutation matrix $\mathbf{P}$, from which we can extract the ordering of the vertices in the maximum clique. This permutation framework fundamentally differs from previous UL approaches. While previous methods encode clique constraints as penalty terms for binary classification, learning node-level probabilities, our framework learns relative node orderings that reveal clique structures. This shift from local classification (binary classification) to global structural relationships (ordering) enables the direct capture of inter-node correlations through permutation patterns.

In this paper, we transform the discrete combinatorial problem into a continuous geometric optimization using Chebyshev-based distances, which allows the model to capture clique structural relationships between nodes. We integrate UL with branch-and-bound algorithms, resulting in improved computational efficiency, especially for large, dense graphs. Our method is able to generalize across sizes, with inference overhead diminishing as graph size increases and outperforming tradi-

tional degree-based ordering. Our approach extends beyond binary classification, revealing how UL can learn fundamental combinatorial structures, suggesting broader applications in CO.

## 2 BACKGROUND

**Branch-and-Bound for Maximum Clique**   The branch-and-bound (BnB) approach has been one of the most effective exact methods for solving MCP, with its performance largely determined by two key components: the vertex selection strategy and the upper bound computation. The algorithm incrementally constructs a clique by recursively selecting vertices while leveraging bounds to prune infeasible branches. A crucial factor in its efficiency is the use of heuristics such as *degree-based vertex ordering* and *coloring-based bounds*, which have been widely adopted in BnB frameworks San Segundo et al. (2011); Tomita and Seki (2003); Konc and Janezic (2007); Li et al. (2017). These techniques—ranging from greedy coloring bounds to efficient vertex selection—have significantly influenced subsequent advances McCreesh and Prosser (2013); San Segundo et al. (2016).

**Unsupervised Learning for Vertex Ordering**   The most relevant UL work on graph ordering for combinatorial optimization is UL for Travelling Salesman Problem (TSP), as explored by Min and Gomes (2023); Min et al. (2023). The goal of TSP is to find the shortest Hamiltonian cycle. Min and Gomes (2023) use a Graph Neural Network (GNN) to construct a soft permutation matrix $\mathbb{T} \in \mathbb{R}^{n \times n}$ and optimize the following loss:

$$\mathcal{L}_{\text{TSP}} = \langle \mathbb{T} \mathbb{V} \mathbb{T}^T, \mathbf{D}_{\text{TSP}} \rangle, \tag{1}$$

where $\mathbb{V}$ represents a Hamiltonian cycle from node $1 \to 2 \to ... \to n \to 1$, and $\mathbf{D}_{\text{TSP}}$ is the distance matrix with self-loop distances set as $\lambda$. Here, a soft permutation matrix is a *doubly stochastic* matrix, meaning that every entry satisfies $\mathbb{T}_{ij} \geq 0$ and both its row and column sums are equal to 1, that is,

$$\sum_{j=1}^{n} \mathbb{T}_{ij} = 1 \quad \text{and} \quad \sum_{i=1}^{n} \mathbb{T}_{ij} = 1 \quad \text{for all } i, j. \tag{2}$$

Such matrices provide a continuous relaxation of discrete permutation matrices, enabling gradient-based optimization while still approximating valid permutations. Since the Hamiltonian cycle constraint holds under any permutation and the order is equivalent with respect to the permutation, optimizing Equation 1 serves as a proxy for solving the TSP, incorporating both the shortest path and Hamiltonian cycle constraints. In other words, the order of vertices in the Hamiltonian cycle is determined by the permutation matrix and we aim to find the one that minimizes the total distance.

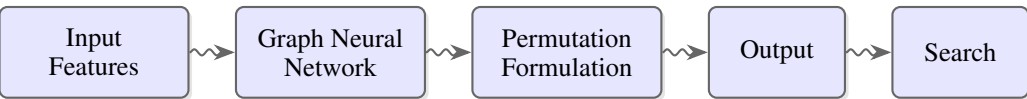

Figure 2: Overview of the unsupervised learning framework for TSP. The model takes graph features as input and processes them through a GNN. The objective is formulated within a permutation framework. The output provides a heat map that guides the subsequent search.

To learn the soft permutation matrix $\mathbb{T}$, Min and Gomes (2023) use a GNN coupled with a Gumbel-Sinkhorn operator. The transformation $\mathbb{T} \mathbb{V} \mathbb{T}^T$ is a heat map representation that guides the subsequent search procedure, as shown in Figure 2.

## 3 MODEL

In our model, the intuition and motivation are straightforward: we aim to learn a good vertex ordering to enhance BnB search performance for MCP. As mentioned in Figure 1, an effective ordering can reveal the hidden clique structure. While most existing search algorithms rely on degree-based vertex ordering, we propose incorporating a *clique-oriented* vertex ordering to guide the search process.

**Learning** We train our model to learn and generate *clique-oriented* ordering following the TSP framework, as illustrated in Figure 2. Our goal is to design a cost matrix $\mathbf{D}_{\text{Clique}}$ analogous to $\mathbf{D}_{\text{TSP}}$ that transforms the discrete constraint satisfaction problem into a continuous geometric optimization. This transformation requires $\mathbf{D}_{\text{Clique}}$ to guide the vertices' reordering process, with the specific aim of clustering vertices in a way that reveals potential clique structures.

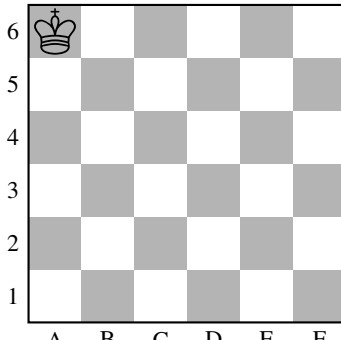

$$
\begin{bmatrix}
0 & 1 & 2 & 3 & 4 & 5 \\
1 & 1 & 2 & 3 & 4 & 5 \\
2 & 2 & 2 & 3 & 4 & 5 \\
3 & 3 & 3 & 3 & 4 & 5 \\
4 & 4 & 4 & 4 & 4 & 5 \\
5 & 5 & 5 & 5 & 5 & 5
\end{bmatrix}
\qquad
\begin{bmatrix}
5 & 4 & 3 & 2 & 1 & 0 \\
4 & 4 & 3 & 2 & 1 & 0 \\
3 & 3 & 3 & 2 & 1 & 0 \\
2 & 2 & 2 & 2 & 1 & 0 \\
1 & 1 & 1 & 1 & 1 & 0 \\
0 & 0 & 0 & 0 & 0 & 0
\end{bmatrix}
$$
$$
C_6 \qquad\qquad\qquad\qquad 5 - C_6
$$

Figure 3: (a): Visualization of a 6x6 chessboard with a king positioned at A6; (b) the Chebyshev distance matrix $C_6$, where each element represents the minimum number of moves required for a king to travel between corresponding squares; (c) $\overline{C_6} = 5 - C_6$, where the elements at top left have larger weights. $C_n[i, j] = \max\{i, j\}$ - 1 and $\overline{C_n}[i, j] = n - 1 - C_n = n - \max\{i, j\}$
.

The key insight of our approach is to reorder vertices such that adjacent pairs are concentrated in specific regions of the matrix. This geometric perspective naturally leads to the Chebyshev distance matrix $C_n$ and its complement $\overline{C_n}$, as illustrated in Figure 3. The Chebyshev distance is defined as the minimum number of moves a king piece requires to traverse a chessboard between two squares. For an $n \times n$ grid, we formalize this distance as $C_n[i, j] = \max\{i, j\} - 1$, with its complement $\overline{C_n}[i, j] = n - \max\{i, j\}$ assigning larger weights to elements in the upper-left region.

The Chebyshev distance matrix is crucial for our model to learn clique structures, as it will naturally guide the optimization to push adjacent vertex pairs (0 in $J - I - A$) toward the upper-left corner, effectively clustering potential clique members together. Furthermore, we can strengthen this geometric intuition by exponentially scaling the distance weights. Specifically, when we set $\mathbf{D}_{\text{Clique}} = (n^2)^{\overline{C_n}}$, minimizing $\mathcal{L}_{\text{Clique}}(P) = \langle \mathbf{P}^T(J - I - A)\mathbf{P}, \mathbf{D}_{\text{Clique}} \rangle$ guarantees convergence to the optimal solution, where $\mathbf{P} \in S_n$ denotes a hard permutation matrix. In practice, we set $\mathbf{D}_{\text{Clique}} = (1 + \epsilon)^{(\overline{C_n} - n/2)}$, where $\epsilon$ is a positive constant. This formulation maintains the exponential weighting scheme and provides better numerical stability.

We train our neural network to minimize the clique-specific objective:

$$
\mathcal{L}_{\text{Clique}} = \langle \mathbb{T}^T(J - I - A)\mathbb{T}, \mathbf{D}_{\text{Clique}} \rangle, \tag{3}
$$

where $\mathbb{T}$ represents a soft permutation matrix.

While this formulation appears similar to the TSP objective in Equation 1, there is a difference in the matrix multiplication order. The TSP formulation uses $\mathbb{T}\mathbb{V}\mathbb{T}^T$, whereas our clique formulation is $\mathbb{T}^T(J - I - A)\mathbb{T}$. This distinction stems from different invariance requirements in the two problems. Let $\mathcal{H}_0 = \mathbb{T}_0\mathbb{V}\mathbb{T}_0^T$ denote the initial heat map of TSP and $\mathbb{T}_0$ is the initial soft permutation matrix. For TSP, the permuted heat map should be equivariant under node reordering. When we apply a permutation matrix $\Pi$ to the original node ordering, our GNN's equivariance ensures $\mathbb{T} = \Pi\mathbb{T}_0$, resulting in a consistently transformed heat map $\Pi\mathcal{H}_0\Pi^T$. Equivariance here means that if we relabel the graph by $\Pi$, the output heat map should relabel in the same way, preserving the structure of the tour under any permutation of node indices.

In contrast, for the maximum clique problem, $\mathbb{T}_0^T(J - I - A)\mathbb{T}_0$ must remain invariant under node reordering. When we apply a permutation $\Pi$, the $J - I - A$ transforms as $J - I - \Pi A \Pi^T$. Due to our GNN's equivariance, $\mathbb{T} = \Pi\mathbb{T}_0$, making $(\Pi\mathbb{T}_0)^T(J - I - \Pi A\Pi^T)(\Pi\mathbb{T}_0)$ equal to the original

$\mathbb{T}_0^T(J - I - A)\mathbb{T}_0$. This invariance is crucial as we aim to reorder adjacent pairs in the upper-left corner, regardless of the initial vertex ordering.

Overall, here we encode the MCP using the same framework as TSP, where discrete combinatorial constraints are transformed into continuous geometric optimization through matrix operations $\mathbb{T}\mathbb{V}\mathbb{T}^T$ and $\mathbb{T}^T(J - I - A)\mathbb{T}$ with distance matrices $\mathbf{D}_{\text{TSP}}$ and $\mathbf{D}_{\text{Clique}}$, respectively.

**Search**  As mentioned, *degree-based vertex ordering* and *coloring-based bound*s are widely adopted in BnB frameworks for solving the MCP Tomita and Seki (2003); Konc and Janezic (2007); Li and Quan (2010); San Segundo et al. (2011); Wu and Hao (2015); Li et al. (2017). Among these methods, MaxCliqueDynKonc and Janezic (2007), an improved version of Tomita et al.'s algorithm Tomita and Seki (2003), is a well-established exact solver that we adopt as our baseline to compare degree-based and UL-based vertex reordering methods. Most BnB methods for the MCP use the MaxCliqueDyn paradigm, which maintains a candidate set of vertices and recursively selects them based on degree or color to construct potential cliques while employing coloring-based bounds for pruning. Extensions such as MaxCliqueDyn+EFL+SCR Li and Quan (2010) integrate failed literal detection and soft clause relaxation but retain MaxCliqueDyn's core structure.

---

**Algorithm 1** MaxCliqueDyn: Maximum Clique Algorithm with Dynamic Upper Bounds

---
**Require:** Graph $G = (V, E)$, candidate set $R \subseteq V$, coloring $C$, depth $level$
**Ensure:** Maximum clique in $G$
1: Initialize $Q \leftarrow \emptyset$, $Q_{max} \leftarrow \emptyset$, $S[level] \leftarrow 0$, $S_{old}[level] \leftarrow 0$      ▷ Cliques and steps
2: $ALL\_STEPS \leftarrow 1$, $T_{limit} \leftarrow 0.025$      ▷ Step counter and threshold
3: Sort $V$ in a non-increasing order with respect to their degrees; color first $\Delta(G)$ vertices $1, \ldots, \Delta(G)$, rest $\Delta(G) + 1$      ▷ Degree-based init coloring
4: **procedure** MAXCLIQUEDYN($R, C, level$)
5:      $S[level] \leftarrow S[level] + S[level - 1] - S_{old}[level]$      ▷ Step count
6:      $S_{old}[level] \leftarrow S[level - 1]$      ▷ Save old count
7:      **while** $R \neq \emptyset$ **do**
8:          $p \leftarrow \arg \max_{v \in R} C(v)$      ▷ Best remaining vertex
9:          $R \leftarrow R \setminus \{p\}$      ▷ Remove vertex
10:          **if** $|Q| + C[index\_of\_p\_in\_R] > |Q_{max}|$ **then**      ▷ Promising bound
11:              $Q \leftarrow Q \cup \{p\}$      ▷ Add to clique
12:              **if** $R \cap \Gamma(p) \neq \emptyset$ **then**      ▷ Has neighbors, where $\Gamma(p)$ denotes the neighborhood of vertex $p$
13:                  **if** $S[level]/ALL\_STEPS < T_{limit}$ **then**      ▷ Near root
14:                      Compute degrees in $G(R \cap \Gamma(p))$      ▷ Better bounds
15:                      Sort $R \cap \Gamma(p)$ by non-increasing degree      ▷ Order by potential
16:                  **end if**
17:                  $C' \leftarrow \text{ColorSort}(R \cap \Gamma(p))$      ▷ Color subgraph
18:                  $S[level] \leftarrow S[level] + 1$      ▷ Count step
19:                  $ALL\_STEPS \leftarrow ALL\_STEPS + 1$      ▷ Update total
20:                  MAXCLIQUEDYN($R \cap \Gamma(p), C', level + 1$)      ▷ Recurse
21:              **else**
22:                  **if** $|Q| > |Q_{max}|$ **then**      ▷ New best
23:                      $Q_{max} \leftarrow Q$      ▷ Update max
24:                  **end if**
25:              **end if**
26:              $Q \leftarrow Q \setminus \{p\}$      ▷ Backtrack
27:          **else**
28:              **return**      ▷ Prune branch
29:          **end if**
30:      **end while**
31: **end procedure**

---

Building upon this representative model, we aim to learn vertex ordering directly from graph data to guide the BnB search, as an alternative to the traditional degree-based ordering used in MaxCliqueDyn.

**MaxCliqueDyn**   MaxCliqueDyn uses dynamic bound adjustment to efficiently solve the MCP. The algorithm maintains two key sets: $Q$ for the current growing clique and $Q_{max}$ for the best solution found. Step counters $S[level]$ and $S_{old}[level]$ track search progress.

The algorithm combines several optimization strategies: non-increasing degree ordering for initial bounds, dynamic step counting for adaptive bound adjustment, and the `ColorSort` algorithm for maintaining vertex ordering properties. By applying bound calculations selectively near the root of the search tree, MaxCliqueDyn achieves significant performance improvements on dense graphs while preserving efficiency on sparse instances Konc and Janezic (2007).

At the beginning, MaxCliqueDyn sorts vertices in non-increasing degree order and assigns the first $\Delta(G)$ vertices colors 1 through $\Delta(G)$ and the remaining vertices color $\Delta(G) + 1$, where $\Delta(G)$ is the maximum degree in $G$. This provides a computationally efficient starting point that supports the algorithm's dynamic bound calculations throughout the search process Tomita and Seki (2003), the algorithm is shown in Algorithm 1. This initial coloring strategy, though simple, establishes a valid starting point for the BnB process. Rather than investing heavily in an optimal initial coloring, it uses this basic coloring that improves automatically through the `ColorSort`. In practice, this simple initial coloring achieves a balance between computation time and reduction in search space Tomita and Seki (2003); Konc and Janezic (2007).

The `ColorSort` procedure plays a crucial role in the BnB framework by providing increasingly refined upper bounds through an approximate graph coloring. Following Konc and Janezic (2007), `ColorSort` first computes $k_{\min} = |Q_{\max}| - |Q| + 1$, which represents the minimum required colors for potential improvements to the current best clique. It then assigns vertices to color classes based on their adjacency relationships, where vertices receiving colors $k < k_{\min}$ are maintained in their original positions, while vertices with colors $k \geq k_{\min}$ are reordered based on their assigned colors, we refer more details to the MaxCliqueDyn paper Konc and Janezic (2007).

**From Soft Permutation $\mathbb{T}$ to Hard Permutation P**   To transform the GNN output into a hard permutation matrix $\mathbf{P}$, we employ a differentiable sorting operation. Specifically, we apply the Gumbel-Sinkhorn operator to the GNN's output, which is a continuous relaxation of the permutation during training while allowing us to obtain a hard permutation matrix during inference through the Hungarian algorithm Mena et al. (2018). This permutation matrix $\mathbf{P}$ is then used to reorder the input vertices, partitioning likely clique nodes together.

In our GNN model, each node has two input features: (1) local density, calculated as the ratio of existing edges to possible edges in the node's neighborhood, and (2) node degree. Our model first generates logits which are transformed by a scaled tanh activation[1]:

$$\mathcal{F} = \alpha \tanh(f_{\text{GNN}}(f_0, A)) \tag{4}$$

where $f_0 \in \mathbb{R}^{n \times 2}$ is the initial feature matrix and $A \in \mathbb{R}^{n \times n}$ is the adjacency matrix. The learned features are transformed into logits which are scaled by $\tanh$ with factor $\alpha$. These scaled logits are then passed through the Gumbel-Sinkhorn operator to build a differentiable approximation of a permutation matrix:

$$\mathbb{T} = \text{GS}(\frac{\mathcal{F} + \gamma \times \text{Gumbel noise}}{\tau}, l), \tag{5}$$

where $\gamma$ is the scale of the Gumbel noise, $\tau$ is the temperature parameter, and $l$ is the number of Sinkhorn iterations. During inference, the logits $\mathcal{F}$ are then directly converted to a hard permutation matrix using the Hungarian algorithm: $\mathbf{P} = \text{Hungarian}(-\frac{\mathcal{F} + \gamma \times \text{Gumbel noise}}{\tau})$.

**Search with Clique-oriented Ordering**   After obtaining the hard permutation matrix $\mathbf{P}$, we reorder the vertices according to this permutation to build what we refer to as the learned *clique-oriented* vertex ordering.

To enhance MaxCliqueDyn's efficiency, we propose replacing the traditional degree-based ordering (line 3 in Algorithm 1) with our clique-oriented vertex ordering learned through UL. To maintain BnB correctness, we follow Konc and Janezic (2007); Tomita and Seki (2003) by coloring the first $\Delta(G)$

---

[1]Our Gumbel–Sinkhorn implementation can use PyTorch's GPU-accelerated tensor operations together with `torch.compile`, which significantly speeds up both the forward optimization and the backward gradient computations in practice.

vertices with unique colors from 1 to $\Delta(G)$ and assigning all remaining vertices color $\Delta(G) + 1$. Since MaxCliqueDyn selects vertices with the highest color label first, this ordering means non-clique vertices are evaluated earlier in the search process, allowing the algorithm to establish good candidate cliques during initial phases. These discovered cliques then serve as effective lower bounds as the search progresses to the potential clique vertices later in the sequence, enabling more aggressive pruning of the search space, thus leading to fewer total steps and faster execution.

In practice, we observe that although the subsequent `ColorSort` procedure in MaxCliqueDyn will modify the initial vertex ordering, the vertices with maximum colors $C(p)$ (which are selected for subsequent procedures) tend to maintain a strong correlation with their initial positions in our clique-oriented ordering. This means that vertices that we initially identified as likely clique members, despite being reordered by `ColorSort` and $R \cap \Gamma(p)$, still tend to be processed later in the search process, where $\Gamma(p)$ denotes the neighborhood of vertex $p$. This delayed processing of potential clique vertices aligns with our original strategy. Thus, the benefits of our clique-oriented ordering persist.

## 4 EXPERIMENTS

**Training** Our dataset consists of Erdős-Rényi (ER) graphs with sizes $n \in \{100, 200\}$ and edge probabilities $p \in \{0.1, 0.2, \ldots, 0.9\}$. For each combination of size and probability, we generate 50,000 training graphs, 10,000 validation graphs, and 10,000 test graphs. We train our GNN using the Adam optimizer with learning rate 0.0001 for 100 epochs per graph configuration. The model architecture uses a two-layer Scattering Attention GNN (SAG) Min et al. (2022) with 6 scattering and 3 low-pass channels, with hidden dimension 128 for $n = 100$ and 256 for $n = 200$. The tanh scale is set to $\alpha = 40$. We conducted experiments using a NVIDIA H100 Graphics Processing Unit (GPU) and an Intel Xeon Gold 6154 Central Processing Unit (CPU).

Table 1: Comparison of MaxCliqueDyn with three orderings (Random, Clique-oriented, and Degree Sort) on random graphs with $n = 10,000$ vertices and varying edge probabilities $p$. We use a 5:1:1 split for training, validation, and test, and each reported value is the average over 10,000 random instances. We report the number of steps and computation time (in seconds) for each algorithm. The Clique-oriented approach includes an additional inference overhead. The maximum clique size $\omega$ is reported in the last column.

| | Random | | Clique-oriented | | | Degree Sort | | |
|------|--------|----------|-----------------|---------|----------------------|-------------|----------|--------|
| $p$ | Steps | Time (s) | Steps | Time | Inference (s) | Steps | Time (s) | $\omega$ |
| 0.1 | **94.25** | 7.799e-5 | 97.22 | 7.290e-5 | *6.357e-5 + 9.424e-4* | 98.45 | 7.321e-5 | 3.962 |
| 0.2 | 110.9 | 9.900e-5 | **107.8** | 9.663e-5 | *6.323e-5 + 1.030e-3* | 108.6 | 9.437e-5 | 5.022 |
| 0.3 | 159.0 | 1.480e-4 | **139.7** | 1.330e-4 | *6.402e-5 + 9.302e-4* | 143.6 | 1.380e-4 | 6.122 |
| 0.4 | 284.7 | 2.565e-4 | **245.7** | 2.192e-4 | *6.379e-5 + 7.107e-4* | 252.4 | 2.296e-4 | 7.514 |
| 0.5 | 535.1 | 5.042e-4 | **434.3** | 3.973e-4 | *6.345e-5 + 8.736e-4* | 456.2 | 4.053e-4 | 9.191 |
| 0.6 | 973.8 | 9.766e-4 | **873.0** | 8.038e-4 | *6.371e-5 + 7.767e-4* | 912.0 | 8.087e-4 | 11.45 |
| 0.7 | 1968 | 1.922e-3 | **1764** | 1.625e-3 | *6.427e-5 + 8.173e-4* | 1792 | 1.612e-3 | 14.65 |
| 0.8 | 4641 | 5.201e-3 | **3904** | 4.200e-3 | *6.550e-5 + 8.862e-4* | 4066 | 4.230e-3 | 19.86 |
| 0.9 | 4870 | 7.752e-3 | **4069** | 6.118e-3 | *6.352e-5 + 1.051e-3* | 4209 | 6.206e-3 | 30.69 |

In our experiments on $n = 100$, we vary the temperature parameter $\tau \in \{1, 2, 3, 4, 5\}$ and the noise scale $\gamma \in \{0.01, 0.02, 0.03, 0.04, 0.05\}$, while fixing $\epsilon = 0.2$ and $l = 20$ for all edge probabilities; on $n = 200$, we use the same variations for $\tau$ and $\gamma$, but set $\epsilon$ to either $0.06$ or $\epsilon = 0.08$, with $l = 10$ for all edge probabilities. We then select the model with fastest inference time on the validation set. The results on the test data are shown in Table 1 and 2.

The best performance is highlighted in bold for the number of steps and underlined for computation time (excluding inference overhead). For $n = 100$, our learned clique-oriented approach achieves the lowest number of steps for all edge probabilities except $p = 0.1$, where random ordering performs marginally better. The reduction in steps becomes more pronounced as edge probability increases, with up to 16.4% fewer steps compared to random ordering at $p = 0.9$. Our learned clique-oriented approach achieves the fastest execution in 7 out of 9 cases, while degree-based ordering performs

Table 2: Comparison of MaxCliqueDyn with three orderings (Random, Clique-oriented, and Degree Sort) on random graphs with $n = 200$ vertices and varying edge probabilities $p$. We use a 5:1:1 split for training, validation, and test, and each reported value is the average over 10,000 random instances. We report the number of steps and computation time (in seconds) for each algorithm. The Clique-oriented approach includes an additional inference overhead. The maximum clique size $\omega$ is reported in the last column.

| | Random | | Clique-oriented | | | Degree Sort | | |
|---|---|---|---|---|---|---|---|---|
| $p$ | Steps | Time (s) | Steps | Time (s) | Inference (s) | Steps | Time (s) | $\omega$ |
| 0.1 | 2.040e+2 | 2.301e-4 | **2.001e+2** | 2.306e-4 | *8.138e-5+5.285e-3* | 2.031e+2 | 2.398e-4 | 4.209 |
| 0.2 | 3.505e+2 | 3.645e-4 | **3.236e+2** | 3.551e-4 | *8.143e-5+5.588e-3* | 3.270e+2 | 3.635e-4 | 5.881 |
| 0.3 | 9.182e+2 | 8.213e-4 | **8.426e+2** | 7.441e-4 | *8.100e-5+3.840e-3* | 8.554e+2 | 7.536e-4 | 7.096 |
| 0.4 | 2.196e+3 | 2.472e-3 | **2.115e+3** | 2.306e-3 | *8.289e-5+5.446e-3* | 2.220e+3 | 2.257e-3 | 8.959 |
| 0.5 | 6.492e+3 | 8.260e-3 | **6.119e+3** | 7.427e-3 | *8.097e-5+5.309e-3* | 6.233e+3 | 7.455e-3 | 11.02 |
| 0.6 | 2.818e+4 | 3.692e-2 | **2.651e+4** | 3.270e-2 | *8.133e-5+4.527e-3* | 2.703e+4 | 3.281e-2 | 13.88 |
| 0.7 | 1.372e+5 | 1.934e-1 | **1.277e+5** | 1.717e-1 | *8.143e-5+6.426e-3* | 1.299e+5 | 1.729e-1 | 18.05 |
| 0.8 | 1.288e+6 | 2.199e+0 | **1.182e+6** | 1.948e+0 | *8.191e-5+6.146e-3* | 1.248e+6 | 2.001e+0 | 25.20 |
| 0.9 | 1.435e+7 | 4.076e+1 | **1.209e+7** | 3.274e+1 | *8.132e-5+5.250e-3* | 1.252e+7 | 3.360e+1 | 41.27 |

best in 2 cases ($p = 0.2$ and $p = 0.7$). The time savings correlate strongly with the reduction in steps. The clique-oriented method does incur an additional inference cost, consisting of two components: GNN inference ($\approx 6.4 \times 10^{-5}$ seconds) and building a hard permutation using the Hungarian algorithm ($\approx 9.0 \times 10^{-4}$ seconds). As the edge probability increases from 0.1 to 0.9, all methods show exponential growth in both steps and computation time. However, the clique-oriented approach maintains its relative advantage, with the benefits becoming more significant for denser graphs. To investigate how our method scales with graph size, we conducted additional experiments on larger graphs with $n = 200$ vertices, with results shown in Table 2.

The performance advantage of the clique-oriented ordering becomes more pronounced as both graph size and density increase. In larger graphs with $n = 200$ vertices, the results are shown in Table 2. Our clique-oriented ordering consistently achieves the lowest number of steps across all edge probabilities, with improvements becoming particularly significant on denser graphs. For sparse graphs ($p = 0.1$), the clique-oriented approach shows a modest improvement, reducing steps by 1.9% compared to random ordering (from $2.040 \times 10^2$ to $2.001 \times 10^2$). This advantage over random ordering grows substantially with edge probability: at $p = 0.6$, steps are reduced by 5.9% (from $2.818 \times 10^4$ to $2.651 \times 10^4$), and at $p = 0.9$, the improvement reaches 15.7% (from $1.435 \times 10^7$ to $1.209 \times 10^7$). Compared with degree-based ordering, the clique-oriented approach reduces 1.9% at $p = 0.6$ and 3.4% at $p = 0.9$. The computation time shows similar trends, with the clique-oriented approach achieving the fastest execution (excluding inference overhead) in 7 out of 9 cases. The time savings become most significant on dense graphs. This substantial improvement more than compensates for the small, constant inference overhead—approximately $8.1 \times 10^{-5}$ seconds for neural network inference plus $5.3 \times 10^{-3}$ seconds for permutation computation.

It should be noted that, at $p = 0.8$ and $p = 0.9$, even when including the inference time overhead, our clique-oriented ordering achieves lower total computation time compared to degree-based sorting. Specifically, on $p = 0.9$, when we run the inference on our CPU (Intel Xeon Gold 6154), it has an average inference time of $\approx 0.04$ seconds, making total execution time for clique-oriented ordering at $p = 0.9$ approximately 32.7 seconds, while degree-sort ordering takes 33.6 seconds, resulting in a 2.6% improvement. This demonstrates that even in a CPU-only environment, our clique-oriented ordering outperforms degree-based ordering. This advantage becomes more pronounced as graph size and density increase, making our method of great practical value. Our results suggest that our UL model successfully captures important structural information that guides more efficient BnB.

## 5 THE LEARNED CLIQUE-ORIENTED ORDERING

To visualize our UL clique-oriented ordering, we select a randomly generated test instance with $n = 200$ vertices and edge probability $p = 0.8$. Figure 4 illustrates different vertex ordering

approaches. The random ordering does not show discernible patterns, making it difficult to identify structural properties. Both the clique-oriented and degree-sorted ordering show a concentration of edges in the upper-left region, but with distinct characteristics. The clique-oriented ordering groups clique members together, revealing dense blocks that correspond to strongly connected subgraphs. In contrast, the degree-sorted ordering emphasizes hub-like nodes but makes clique structures less distinguishable, resulting in less distinct dense blocks.

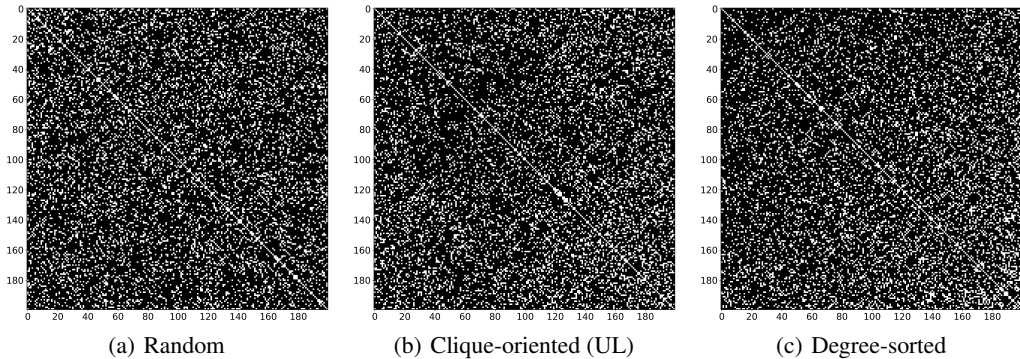

(a) Random          (b) Clique-oriented (UL)          (c) Degree-sorted

Figure 4: Adjacency matrix visualization of the graph: (a) random ordering, (b) clique-oriented ordering, and (c) matrix sorted by non-increasing degree.

Figure 5 shows adjacency matrices for the first 50 vertices, highlighting cliques of size $\geq 5$. The random ordering (a) exhibits minimal clique structures, while both clique-oriented (b) and degree-sorted (c) orderings effectively cluster vertices belonging to cliques. The clique-oriented ordering demonstrates better clique identification, revealing 7 distinct cliques compared to 6 in the degree-sorted ordering, with cliques positioned closer to the upper-left corner. This validates the effectiveness of our UL approach in revealing inherent clique structures through reordering.

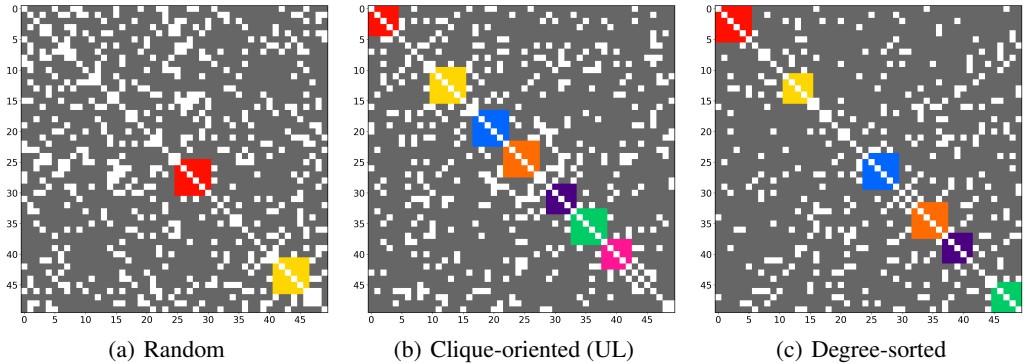

(a) Random          (b) Clique-oriented (UL)          (c) Degree-sorted

Figure 5: Adjacency matrix of the first 50 nodes of the graph: (a) random ordering, (b) clique-oriented ordering, and (c) matrix sorted by non-increasing degree.

These visual results clearly reflect the goal of our method. The Chebyshev-inspired distance matrix $\mathbf{D}_{\text{Clique}}$ encourages the model to pull likely clique-forming vertex pairs toward the top-left corner, and during training it naturally learns to group densely connected vertices together. The dense blocks that appear in the clique-oriented ordering are therefore a direct outcome of this objective, showing that the model is uncovering meaningful structure rather than arranging vertices arbitrarily. This reordering also benefits exact MCP solvers: when clique candidates appear early, the solver can identify a large clique sooner, tighten the initial lower bound, and prune the search space more effectively. In contrast, degree-based sorting provides no explicit structural bias toward clique formation and often scatters true clique vertices.

Although the clique-oriented (UL) ordering and degree-sorted ordering may look similar at a global level, this is expected because both pull well-connected vertices toward the top-left region. The

important differences appear in the finer structure. The UL ordering produces sharper, more coherent dense blocks that align more closely with actual clique memberships, as shown in the zoomed-in view of the first 50 nodes, as shown in Figure 5. Degree sorting tends to group high-degree vertices that are not necessarily part of the same clique, leading to more diffuse patterns. This subtle but consistent sharpening of clique-related regions in the UL ordering is what ultimately gives it a performance advantage, allowing the solver to focus on promising regions of the search space earlier and prune more aggressively.

## 6 CONCLUSION

In this paper, we demonstrate that UL can be used for reordering, where the resulting reordering reveals underlying combinatorial structures. Instead of formulating the MCP as a binary classification problem, we encode it using a permutation framework. This approach enables us to learn the ordering of vertices directly, rather than making binary decisions. After decoding the model's output, the clique structures are naturally revealed. Importantly, reordering and binary classification approaches are not mutually exclusive: while binary classification focuses on direct yes/no decisions about whether nodes belong to the solution, reordering provides a complementary perspective by uncovering the inherent structural relationships between nodes. By integrating both approaches, we can leverage their respective strengths: binary classification's explicit decision-making and reordering's ability to capture structural patterns.

Our experiments with MaxCliqueDyn demonstrated that traditional degree-based ordering in BnB can be improved through UL approaches. As graph size and density increase, our inference overhead becomes proportionally smaller in the total execution time. Notably, on the largest, densest graphs ($n = 200, p = 0.9$), our approach outperforms degree-based ordering even when accounting for inference time. This demonstrates the practical viability of our UL method, particularly for challenging instances. Given that MaxCliqueDyn is a representative BnB algorithm and degree-based ordering is widely used in most exact clique solvers, these results suggest the potential for improving exact solvers through learned ordering strategies. In this paper, we only replaced the initial degree-based ordering with our learned clique-oriented ordering. There remain many promising directions for further incorporating clique-oriented ordering into existing algorithms, such as exploring deeper integration of learned clique-oriented methods throughout the search process, beyond just initialization.

**Sensitivity to Hyperparameters.** We examined how different hyperparameters affect performance and found the model to be generally robust. Increasing the GNN from 2 to 3 layers still improves MaxCliqueDyn, achieving $1.232 \times 10^7$ steps and 33.01 seconds on ($n = 200, p = 0.9$) compared to $1.252 \times 10^7$ steps and 33.60 seconds. Gumbel noise has only a minor effect, with noise levels of 0.01 and 0.05 both yielding about a 2.5% runtime reduction. The most sensitive component is the Chebyshev-based matrix $\mathbf{D}_{\text{Clique}}$. Our choice $\mathbf{D}_{\text{Clique}} = (1 + \varepsilon)^{(C_n - n/2)}$ works well for small $\varepsilon$, while larger values (e.g., $\varepsilon = 0.5$) cause numerical instability and degrade performance. This can be mitigated by using smaller $\varepsilon$ or a slower-growing polynomial form, as long as weights increase toward the top-left to promote clique formation.

**Discussion and Future Work.** In this work, we demonstrate that an unsupervised learning-based approach can already improve a widely used exact solver through its initial ordering alone. The key point is that our ordering is learned, not hand-crafted, allowing the model to automatically discover structural patterns that traditional heuristics may not be able to capture.

We focus on learning effective initial vertex orderings for MaxCliqueDyn, but the idea is not tied to this solver. Since most exact MCP frameworks depend on an initial ordering to guide branching and pruning, our approach can naturally support many solvers. An important advantage is that the method is fully *unsupervised*, requiring no labels and making it cheap to train and easy to deploy. A supervised variant that learns from high-quality orderings could yield even stronger improvements. Node ordering remains crucial in modern exact MCP algorithms, and a good ordering can serve as a simple yet effective preprocessing step for recent methods as well Li et al. (2013); Li and Quan (2010); San Segundo et al. (2019). Exploring deeper integration between learning and search is a promising direction for future work.

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

## A    COMPUTE THE HARD PERMUTATION

In our implementation, we use `scipy.optimize.linear_sum_assignment` to compute the final hard permutation. We also tested the open-source CUDA batched assignment solver from Karpukhin et al. (2024), which substantially speeds up hard-permutation decoding when using large batch sizes.

## B    GENERALIZATION

To investigate our model's capability to handle varying graph dimensions, we employ a zero-padding strategy for size generalization. Given a graph with $n = 190$ nodes and edge probability $p = 0.9$, we pad it with 10 dummy nodes to match our training dimension. To ensure similar edge density between training and testing graphs after padding, we train the model on ER random graphs with $n = 200$ nodes and edge probability $p = 0.81$. Specifically, these dummy nodes have zero feature vectors, and their corresponding entries in the adjacency matrix are also set to zero, making them isolated nodes. This padding strategy provides a general approach to handle size differences: smaller graphs can be padded to the larger sizes, enabling our model to process arbitrary sizes.

Table 3: Comparison of MaxCliqueDyn with three orderings (Random, Generalized Clique-oriented, and Degree Sort) on random graphs with $n = 190$ vertices and edge probabilities $p = 0.9$. For each algorithm, we report the number of steps taken and computation time in seconds. The size of the largest clique found $\omega$ is shown in the rightmost column.

| | Random | | Generalized Clique-oriented | | | Degree Sort | | |
|---|---|---|---|---|---|---|---|---|
| $p$ | Steps | Time (s) | Steps | Time (s) | Inference (s) | Steps | Time (s) | $\omega$ |
| 0.9 | 7.145e+6 | 1.923e+1 | **5.656e+6** | 1.476e+1 | *8.191e-5+6.146e-3* | 5.886e+6 | 1.523e+1 | 40.46 |

We use the same training method described in Section 4 and our results are shown in Table 3. The generalized clique-oriented model performs effectively, requiring $5.656 \times 10^6$ steps and 14.76 seconds to find the maximum clique in an ER random graph with $n = 190$ and $p = 0.9$. This result outperforms both the random algorithm ($7.145 \times 10^6$ steps, 19.23 seconds) and the degree sort method ($5.886 \times 10^6$ steps, 15.23 seconds). The additional inference overhead of our method (approximately 6.23 milliseconds) is negligible compared to the overall computation time, demonstrating that our generalized approach maintains efficiency while handling different sizes.

## C    PROOF

The following proof discusses the connection between the Chebyshev distance complement $\overline{C_n}$ in exponential form and the maximum clique problem. Specifically,

**Lemma 1.** *When* $\mathbf{D}_{Clique} = (n^2)^{\overline{C_n}}$ *with* $\overline{C_n}[i,j] = n - \max(i,j)$, *minimizing* $\mathcal{L}_{clique}(P) = \langle P^T(J - I - A)P, \mathbf{D}_{Clique} \rangle$ *yields the maximum clique.*

*Proof.* Let $G = (V, E)$ be an undirected graph with adjacency matrix $A$. The matrix $J - I - A$ represents non-adjacent vertex pairs, where $J \in \mathbb{R}^{n \times n}$ is the all-ones matrix and $I \in \mathbb{R}^{n \times n}$ is the identity matrix.

Let $\omega$ be the size of the maximum clique in $G$. We will show that any permutation matrix that minimizes $\mathcal{L}_{\text{clique}}(P)$ must place the maximum clique in the first $\omega$ positions.

Let $P_1$ be a permutation matrix that places a maximum clique of size $\omega$ in the first $\omega$ positions. The corresponding cost is:

$$\mathcal{L}_{\text{clique}}(P_1) = \sum_{i=1}^{n} \sum_{j=1}^{n} [P_1^T(J - I - A)P_1]_{ij} \cdot (n^2)^{n - \max(i,j)} \tag{6}$$

Since the first $\omega$ vertices form a clique, we have $[P_1^T(J - I - A)P_1]_{ij} = 0$ for all $1 \leq i, j \leq \omega$. Non-adjacent vertex pairs can only exist in positions where at least one index exceeds $\omega$. For these positions, we have $n - \max(i,j) \leq n - (\omega + 1) = n - \omega - 1$. Therefore:

$$\mathcal{L}_{\text{clique}}(P_1) \leq \sum_{\substack{i,j \\ \max(i,j)>\omega}} [P_1^T(J - I - A)P_1]_{ij} \cdot (n^2)^{n-\max(i,j)} \tag{7}$$

$$\leq \sum_{\substack{i,j \\ \max(i,j)>\omega}} (n^2)^{n-\max(i,j)} \tag{8}$$

$$\leq (n^2 - \omega^2) \cdot (n^2)^{n-\omega-1} \tag{9}$$

Now, let $P_2$ be any permutation matrix that does not place the maximum clique in the first $\omega$ positions. Then at least one vertex from the maximum clique must be placed at position $\omega + 1$ or beyond, and at least one non-clique vertex must be placed among the first $\omega$ positions.

Since the first $\omega$ positions cannot contain only clique vertices, there must exist at least one pair of vertices in the first $\omega$ positions that are not adjacent. This non-adjacent pair contributes a value of 1 to $[P_2^T(J - I - A)P_2]_{ij}$ where $\max(i,j) \leq \omega$. The corresponding weight is at least $(n^2)^{n-\omega}$. Therefore:

$$\mathcal{L}_{\text{clique}}(P_2) \geq (n^2)^{n-\omega} \tag{10}$$

We can now directly compare the bounds:

$$\frac{\mathcal{L}_{\text{clique}}(P_1)}{\mathcal{L}_{\text{clique}}(P_2)} \leq \frac{(n^2 - \omega^2) \cdot (n^2)^{n-\omega-1}}{(n^2)^{n-\omega}} \tag{11}$$

$$= \frac{n^2 - \omega^2}{n^2} \tag{12}$$

$$= 1 - \frac{\omega^2}{n^2} < 1 \tag{13}$$

This implies $\mathcal{L}_{\text{clique}}(P_1) < \mathcal{L}_{\text{clique}}(P_2)$ for any permutation $P_2$ that does not place the maximum clique in the first $\omega$ positions. Therefore, any permutation matrix that minimizes $\mathcal{L}_{\text{clique}}(P)$ must place the maximum clique in the first $\omega$ positions. $\qquad\square$

