# OpenReview forum: "Unsupervised Ordering for Maximum Clique"
_ICLR.cc/2026/Conference — Submitted to ICLR 2026_

### Official Review · Reviewer_rmLD · 2025-10-21

**Soundness:** 3
**Presentation:** 2
**Contribution:** 2
**Rating:** 6
**Confidence:** 4

**Summary:**

This paper presents a novel unsupervised learning approach for the maximum clique problem. Instead of formulating it as a typical binary classification task, the authors frame the problem as one of learning a permutation-based vertex ordering. This learned ordering is then integrated into a classic BnB solver, replacing the traditional degree-based heuristic. The goal is to guide the search more effectively, reduce the number of computational steps, and improve overall efficiency.

**Strengths:**

1. By moving away from binary classification and recasting the MCP as a permutation-based ordering problem, the authors provide a new and insightful way to apply UL to this combinatorial problem.

2. The approach is practical as it is designed to be integrated with classic BnB solvers.

**Weaknesses:**

1. The reported reduction in computation time appears marginal. For example, in the best-case scenario on the most difficult graphs (n=200, p=0.9), the total time is reduced from 33.6s to 32.7s, which is not a substantial or fundamental improvement. Furthermore, the paper only reports the average time over 100 random instances. The average can be skewed by outliers.

2. The experimental evaluation relies entirely on randomly generated graphs. No real-world datasets are used for testing. It is unclear if the ordering strategy learned on random graphs will generalize to real-world graphs.

**Questions:**

1. Could the authors provide a more detailed explanation of Figures 4 and 5?

2. The paper selected MaxCliqueDyn, a 2007 algorithm, as its baseline. While the paper cites several newer, more advanced methods, it only justifies MaxCliqueDyn as representative. Can the authors comment on whether their learned ordering approach is likely to provide a similar speedup for these more modern, state-of-the-art solvers, which may already incorporate more advanced mechanisms than the baseline's simple initial degree sort?

**Details Of Ethics Concerns:**

N/A.

---

> ### Author Response · Authors · 2025-11-19
> **Response to Reviewer rmLD**
>
> Dear Reviewer rmLD, thank you for your comments. Below is our detailed response to your questions.
>
> ---
>
> *Weakness 1: reports the average time over 100 random instances and time reduction.*
>
> Answer: We are deeply sorry for the typo. Our test set contains 10,000 instances, and our method uses 50,000 for training and 10,000 each for validation and testing, resulting in a 5:1:1 split. This is detailed in the **Experiment** section. we already revise and fix it. In the best-case scenario on the most challenging graphs (n = 200, p = 0.9), the total runtime decreases from 33.6s to 32.7s, which corresponds to a 2.5% reduction. This improvement is meaningful because our approach is fully unsupervised and works as a simple plug-in component.
>
> -----
>
> *Weakness 2: The experimental evaluation relies entirely on randomly generated graphs. No real-world datasets are used for testing.*
>
> Answer: We further test our method on several instances taken from the DIMACS benchmark set. The selected instances include:
>
> - brock200_4.clq
> - brock200_3.clq
> - brock200_2.clq
> - brock200_1.clq
> - c-fat200-5.clq
> - c-fat200-2.clq
> - c-fat200-1.clq
> - san200_0.9_3.clq
> - san200_0.9_2.clq
> - san200_0.9_1.clq
> - san200_0.7_2.clq
> - san200_0.7_1.clq
> - sanr200_0.9.clq
> - sanr200_0.7.clq
>
> ### Baseline Results
>
> - Average steps: 498,405
> - Average time: 1.306 seconds
>
> ### Results Using Our UL Method
>
> After training our UL-based model on instances with similar density, we apply it to the same DIMACS benchmark instances. The results are:
>
> - Average steps: 489,350
> - Average time: 1.265 seconds
>
> These results suggest that our model generalizes well to unseen, real-world instances.
>
> ----
>
> *Question 1:  Could the authors provide a more detailed explanation of Figures 4 and 5?*
>
> Answer: Figure 4 shows the full adjacency matrix of a graph under three different vertex orderings: (a) a random ordering, (b) our clique-oriented (UL) ordering, and (c) a degree-sorted ordering. Because this example is a dense Erdős–Rényi graph, the global structure appears mostly uniform in all three matrices. However, even at this global scale, the clique-oriented ordering produces a slightly more concentrated pattern of edges toward the upper-left region, indicating that it is beginning to cluster mutually adjacent vertices. In contrast, the random ordering exhibits no structure, and the degree-sorted ordering mostly reflects degree variation rather than clique structure. This figure illustrates that, while differences are subtle at the global level, our method already begins to reorganize the matrix toward a clique-friendly layout.
>
> Figure 5 focuses on the first 50 nodes under the same three orderings, which makes the differences much more visible. In the random ordering (a), edges appear scattered with no coherent pattern, and vertices belonging to the same clique are dispersed. In the clique-oriented ordering (b), multiple dense diagonal blocks emerge, shown by the highlighted colored regions; these blocks correspond to subsets of vertices that are highly connected to each other, which is exactly the structure exploited by exact Maximum Clique algorithms. In the degree-sorted ordering (c), some block structure is visible due to high-degree vertices clustering together, but the cliques are not grouped as cleanly or distinctly as in our learned ordering. Overall, Figure 5 demonstrates more clearly that our method succeeds in grouping clique-related vertices, producing a matrix layout that is more favorable for branch-and-bound pruning and early discovery of large cliques.
>
> We have revised the manuscript to provide a more detailed explanation of Figures 4 and 5.
>
> -----

---

> > ### Author Response · Authors · 2025-11-19
> > **Response to Reviewer rmLD**
> >
> > *Question 2:  The paper selected MaxCliqueDyn, a 2007 algorithm, as its baseline. While the paper cites several newer, more advanced methods, it only justifies MaxCliqueDyn as representative. Can the authors comment on whether their learned ordering approach is likely to provide a similar speedup for these more modern, state-of-the-art solvers, which may already incorporate more advanced mechanisms than the baseline's simple initial degree sort?*
> >
> > Answer: We selected MaxCliqueDyn as our baseline because it is a widely used  branch-and-bound framework whose performance is strongly influenced by the initial vertex ordering. More importantly, our contribution is orthogonal to the internal pruning or bounding mechanisms used in modern exact solvers.
> >
> > In fact, since newer algorithms often employ more powerful upper-bound computations or dynamic pruning, they may benefit even more from high-quality initial orderings, because a good ordering can amplify the effect of these mechanisms. Regarding speed,  our method is only a lightweight preprocessing step that computes a single permutation before the solver begins. The solver then runs exactly as before, but on a better-ordered graph. The cost of generating this ordering is negligible compared to the solver’s overall search time, especially for difficult instances.
> >
> >
> > To address this in future work, we plan to integrate our ordering into more recent frameworks such as MaxSAT-INC and ICTAI’10 [1,2], as well as other modern branch-and-bound solvers. Because our approach is a simple preprocessing step that does not alter the solver’s internal logic, we believe it can be easily adopted and is likely to provide similar speedups in these more advanced settings.
> >
> > The most important point is that our approach is **purely unsupervised**, which requires no labels, making it extremely cheap to train and deploy. If one were to relax this constraint and adopt a supervised learning setup, using known maximum cliques as training targets. It is likely that even stronger and more specialized orderings could be learned, further improving the performance of advanced MCP solvers.
> >
> > **We have added a discussion and future work section in our revised manuscript.**
> >
> > [1] Li, Chu-Min, Zhiwen Fang, and Ke Xu. "Combining MaxSAT reasoning and incremental upper bound for the maximum clique problem." 2013 IEEE 25th international conference on tools with artificial intelligence. IEEE, 2013.
> >
> > [2] Li, Chu-Min, and Zhe Quan. "Combining graph structure exploitation and propositional reasoning for the maximum clique problem." 2010 22nd IEEE international conference on tools with artificial intelligence. Vol. 1. IEEE, 2010.

---

### Official Review · Reviewer_bCdK · 2025-10-28

**Soundness:** 3
**Presentation:** 3
**Contribution:** 3
**Rating:** 8
**Confidence:** 3

**Summary:**

In the maximum clique problem, the goal is to find a largest vertex set, such that the induced sub-graph is complete. By complementing the edges this is equivalent to the maximum independent set problem. The paper studies the later actually, which allows them not to have to distinguish diagonal and non-diagonal entries in the adjacency matrix. It is an NP-hard problem, which can be solved exactly by a branch and bound search. It consists of the exploration of a search tree, where every node of the tree is associated to a vertex of the graph, and there are two descendants, one where the vertex is included in the solution, and one where it is not. Deciding on an order on the vertices can influence the running time, where at level i of the tree, the i-th vertex in this order is chosen. A similar approach was proposed for the traveling salesman problem in the past.

Concretely, a soft-permutation T is selected which minimizes the inner product of the resulting adjacency matrix with some weight matrix. This weight matrix consists of exponential decreasing weights depending on the Lmax coordinates norm. Ideally a solution to this problem will result in an adjacency matrix an all zero square in the top left corner of the adjacency matrix, representing an independent set.

This soft permutation matrix T is found with a graph neural network. Then some technique is used to transform it into a hard permutation matrix, using a Gumbel-Sinkhorn operator.

Then experiments are conducted on Erdos-Rényi random graphs.

**Strengths:**

The paper contributes to an important central problem in combinatorial optimization. It shows that unsupervised learning can help reducing the running time of exact algorithms. This is a direction, which the optimization community has to explore these days.

**Weaknesses:**

I would have liked to see experiments on the DIMACS benchmark set.

**Questions:**

I have difficulties to judge the work, since it is far from my expertise. I don't know what a soft permutation matrix, I guess it is a stochastic matrix.

You could mention in the introduction that it maximum clique is hard to approximate. No O(n^1-epsilon)-approximation ratio is possible if P != NP.

page 4 line 194. I don't know the word equivariant.

---

> ### Author Response · Authors · 2025-11-19
> **Response to Reviewer bCdK**
>
> Dear Reviewer bCdK, thank you for your comments. Below is our detailed response to your questions.
>
> -----
>
> *Weakness 1: I would have liked to see experiments on the DIMACS benchmark set.*
>
> Answer: We further test our method on several instances taken from the DIMACS benchmark set. The selected instances include:
>
> - brock200_4.clq
> - brock200_3.clq
> - brock200_2.clq
> - brock200_1.clq
> - c-fat200-5.clq
> - c-fat200-2.clq
> - c-fat200-1.clq
> - san200_0.9_3.clq
> - san200_0.9_2.clq
> - san200_0.9_1.clq
> - san200_0.7_2.clq
> - san200_0.7_1.clq
> - sanr200_0.9.clq
> - sanr200_0.7.clq
>
> ### Baseline Results
>
> - Average steps: 498,405
> - Average time: 1.306 seconds
>
> ### Results Using Our UL Method
>
> After training our UL-based model on instances with similar density, we apply it to the same DIMACS benchmark instances. The results are:
>
> - Average steps: 489,350
> - Average time: 1.265 seconds
>
> ----
>
> *Question 1: ... what a soft permutation matrix, I guess it is a stochastic matrix.*
>
> Answer: Yes, a soft permutation matrix is one whose rows and columns each sum to 1 and whose entries are all positive. Such matrices are often referred to as doubly stochastic, meaning that both the row sums and column sums are constrained to be 1. **We have revised the manuscript to include a clearer description of the soft permutation matrix.**
>
> ----
>
> *Question 2: .You could mention in the introduction that it maximum clique is hard to approximate. No $O(n^{1-\epsilon})$-approximation ratio is possible if P != NP.*
>
> Answer: Thank you for the suggestion. We have incorporate this point into the introduction by noting that the Maximum Clique Problem is not only NP-hard but also hard to approximate—specifically, that no $O(n^{1-\varepsilon}))$-approximation is possible unless P=NP. T his addition will help emphasize the inherent difficulty of the problem and the motivation for developing stronger heuristics and exact methods.
>
> -----
>
> *Question 3: definition of equivariant*
>
> Answer: “Equivariant” means that if a transformation is applied to the input—such as permuting or relabeling the vertices of a graph—the model’s output transforms in the same way. This contrasts with “invariant,” where the output remains unchanged under such transformations. At the end of page 4, the cited work refers to permutation-equivariant models used in TSP, but in our setting we require invariance, not equivariance, because the final objective (the learned vertex ordering) should not change simply due to a relabeling of the input graph.
>
> In short, equivariance is background knowledge relevant to TSP models, but it is not used in our paper.
>
> **We have revised the manuscript to clarify this definition.**

---

### Official Review · Reviewer_NoFh · 2025-10-31

**Soundness:** 3
**Presentation:** 3
**Contribution:** 2
**Rating:** 4
**Confidence:** 4

**Summary:**

This paper introduces an unsupervised learning approach for the maximum clique problem (MCP) by framing the task as one of discovering informative vertex orderings, rather than binary node classifications. The method leverages a permutation-based geometric formulation, where the combinatorial constraints of MCP are translated into matrix relationships using Chebyshev distances. The learned ordering is then incorporated into the branch-and-bound (BnB) search in place of traditional degree-based vertex ordering. Experimental results on synthetic Erdős-Rényi graphs show reductions in search steps and computation time.

**Strengths:**

1. The paper proposes a clear shift in how unsupervised learning is used for the maximum clique problem: framing MCP as a permutation (ordering) learning task rather than standard vertex-wise classification, resulting in a fundamentally different optimization strategy.

2. The methodology is rigorously constructed, with a detailed introduction of the Chebyshev distance matrix and its role in aligning matrix structure with optimal clique placement. The distinction from permutation-based approaches for problems like TSP is carefully described.

3. Empirical results are detailed and convincing: Table 1 and Table 2 show that for both ( n=100 ) and ( n=200 ) graphs, the clique-oriented ordering consistently beats or matches degree-based and random orderings in steps and computation time for most edge probabilities, and the benefit is especially clear for denser graphs.

**Weaknesses:**

1. All experiments are on synthetic Erdős-Rényi graphs. MCP is notoriously harder on structured or sparse graphs and in real-world network scenarios, where the correlation between degree and clique participation can be very weak or non-monotonic.

2. The only empirical baselines are random ordering and classical degree-based sorting. Actually, graph ordering have been extensively studied in the litearture, and many ordering methods have been explored, such as "Can Graph Reordering Speed Up Graph Neural Network Training? An Experimental Study"

3. The method is only used to initialize MaxCliqueDyn ordering; most of the potential for “deep integration” promised in the conclusion is left unexplored, so the actual scientific contribution lies in proposing a learning-based initial ordering. This is a modest advance unless future work (or additional experiments) make a stronger case for ongoing integration.

**Questions:**

1. How does the proposed approach perform on structured or real-world benchmarks beyond synthetic ER graphs? What are the limitations when applied to graphs with strong local patterns, inhomogeneous degree distributions, or scale-free structure?

2. What is the sensitivity of performance to key architectural choices such as GNN layers, the size of the hidden layer, the value of ( $\alpha$ ), Gumbel/Sinkhorn parameters, and the initial feature design? Have alternative feature sets or architectural variants (e.g., using VNN or transformer encoders) been tested?

3. Are there scenarios where the learned ordering “fails” (e.g., is worse than degree-based), and what are the characteristics of these failures?

---

> ### Author Response · Authors · 2025-11-19
> **Response to Reviewer NoFh**
>
> Dear Reviewer NoFh, thank you for your comments. Below is our detailed response to your questions.
>
> ------
>
> *Weakness 1: Weakness: All experiments are on synthetic Erdős-Rényi graphs. MCP is notoriously harder on structured or sparse graphs and in real-world network scenarios, where the correlation between degree and clique participation can be very weak or non-monotonic.*
>
> Answer: Our ER examples cover a wide range of densities, with p=0.1 to 0.9. Although it is sometimes noted that “the MCP is harder on sparse graphs,” we believe $\underline{\text{the opposite is generally true for unstructured random graphs}}$: the MCP is $\underline{\text{typically hardest on dense graphs}}$, because large cliques are common and pruning becomes ineffective, though certain structured sparse graphs may also be challenging.
>
> To provide a broader evaluation beyond the Erdős–Rényi (ER) family, we additionally tested our method on the following graph models:
>
> - Brabási–Albert (BA) graphs with parameters ( n = 200 , m = 20 )
> - Watts–Strogatz (WS) graphs with parameters ( n = 20, k = 20 , p = 1 )
>
> On these non-ER graph types, our learned clique-oriented ordering continued to provide improvements over traditional degree-based ordering within the MaxCliqueDyn framework. We observed:
>
> - 1.05% reduction in computation time on Barabási–Albert graphs
> - 2.98% reduction in computation time on Watts–Strogatz graphs
>
> These results demonstrate that our method generalizes well across different graph topologies and maintains efficiency improvements beyond ER graphs.
>
> We further test our method on several instances taken from the DIMACS benchmark set. The selected instances include:
>
> - brock200_4.clq
> - brock200_3.clq
> - brock200_2.clq
> - brock200_1.clq
> - c-fat200-5.clq
> - c-fat200-2.clq
> - c-fat200-1.clq
> - san200_0.9_3.clq
> - san200_0.9_2.clq
> - san200_0.9_1.clq
> - san200_0.7_2.clq
> - san200_0.7_1.clq
> - sanr200_0.9.clq
> - sanr200_0.7.clq
>
> ### Baseline Results
>
> - Average steps: 498,405
> - Average time: 1.306 seconds
>
> ### Results Using Our UL Method
>
> After training our UL-based model on instances with similar density, we apply it to the same DIMACS benchmark instances. The results are:
>
> - Average steps: 489,350
> - Average time: 1.265 seconds
>
> These results suggest that our model generalizes well to unseen, real-world instances.
>
> ------
>
> *Weakness 2:  The only empirical baselines are random ordering and classical degree-based sorting. Actually, graph ordering have been extensively studied in the litearture, and many ordering methods have been explored, such as "Can Graph Reordering Speed Up Graph Neural Network Training? An Experimental Study"*
>
> Thank you for this observation. We agree that graph reordering is a foundational topic with a substantial body of related work, and we will expand our discussion to reflect this broader context and add more related paper in the introduction section.  We also want to highlight that our formulation is designed so that the matrix $P^T A P$  is invariant under graph relabeling (see end of page 4). Invariance is required in our setting because the final objective, namely the learned vertex ordering, should not change simply because the input graph is permuted.
>
> **In the revised manuscript, we have included a more detailed discussion of graph reordering methods studied in the literature, including those explored in the GNN  and maximum clique solver community.**
>
> -------
>
>
> *Weakness 3: The method is only used to initialize MaxCliqueDyn ordering; most of the potential for “deep integration” promised in the conclusion is left unexplored, so the actual scientific contribution lies in proposing a learning-based initial ordering. This is a modest advance unless future work (or additional experiments) make a stronger case for ongoing integration.*
>
> In the current version of the paper, we focus on using our method to initialize the ordering in MaxCliqueDyn, which already yields a
> 2.5 \% runtime reduction for n=200, p=0.9. Our goal is to show that a purely unsupervised approach, which requires no labels, can already improve a widely used exact solver through initial ordering alone. We believe that achieving measurable gains with such an unsupervised method is already a meaningful contribution.
>
> While our current scope is limited to initialization in MaxCliqueDyn, a supervised setup using known maximum cliques could potentially learn even stronger orderings. We will clarify this scope more explicitly and refine the discussion in the conclusion. We will also add more discussion on how our approach may be integrated into more advanced solvers, where better orderings can have an even larger impact. Although building a thorough and generic integration for all solvers is beyond the scope of the present paper, we will emphasize that this is a promising direction for future work.
>
> **We have added a discussion and future work section in our revised manuscript.**

---

> > ### Author Response · Authors · 2025-11-19
> > **Response to Reviewer NoFh**
> >
> > *Question 1: How does the proposed approach perform on structured or real-world benchmarks beyond synthetic ER graphs? What are the limitations when applied to graphs with strong local patterns, inhomogeneous degree distributions, or scale-free structure?*
> >
> > Answer: We refer to the answer in the beginning.
> >
> > ---
> > *Question 2: What is the sensitivity of performance to key architectural choices such as GNN layers, the size of the hidden layer, the value of (alpha), Gumbel/Sinkhorn parameters, and the initial feature design? Have alternative feature sets or architectural variants (e.g., using VNN or transformer encoders) been tested?*
> >
> > Answer: We also evaluate how the model’s performance changes under different parameter settings. When modifying the GNN architecture, we find that the method consistently remains competitive with MaxCliqueDyn, e.g. when we change the  GNN from 2 layers to 3 layers, our learned ordering still yields 1.232e+7  steps on average and 33.01 seconds on n=200, p=0.9, while the MaxCliqueDyn takes 1.252e+7 steps and 33.60 seconds.  In our ablation study, we find that the Gumbel noise has only a minor effect on the results. For example, for the setting with $n = 200$ and $p = 0.9$, using a Sinkhorn temperature of $4.0$ with noise levels of $0.01$ and $0.05$ both lead to roughly a 2.5 \% reduction in total runtime. The most influential parameter is the weighting in the Chebyshev-based matrix $\mathcal{D}_{\mathrm{clique}}$. see next question.
> >
> > -----
> >
> >
> > *Question 3: Are there scenarios where the learned ordering “fails” (e.g., is worse than degree-based), and what are the characteristics of these failures?*
> >
> > Answer: In our paper, we set
> >
> > $$
> > \mathcal{D}_{\mathrm{clique}} = (1 + \epsilon)^{(C_n - n/2)},
> > $$
> >
> > where $\epsilon$ is a positive constant. In our experiments, $\epsilon$ is chosen to be relatively small. When we tried a larger value, for example, $\epsilon = 0.5$ on graphs of size $100$, the model became unstable during training and sometimes produced results worse than simple degree-based sorting. We believe this is due to numerical issues caused by the rapid exponential growth in $\mathcal{D}_{\mathrm{clique}}$.
> >
> > This problem can be mitigated by using a smaller $\epsilon$, or by redefining $\mathcal{D}_{\mathrm{clique}}$ to grow more slowly, possibly using a polynomial instead of an exponential form, as long as the weighting still emphasizes the top-left region to encourage clique formation. For instance, one could consider something like $(\bar{C}_n - n/2)^2$ or another polynomial weighting.
> >
> > In practice, our current choice works well, and we believe that any formulation that increases weights toward the top left while avoiding numerical explosion should be sufficient.

---

### Official Review · Reviewer_TboW · 2025-10-31

**Soundness:** 3
**Presentation:** 3
**Contribution:** 3
**Rating:** 6
**Confidence:** 4

**Summary:**

This paper proposes an unsupervised learning framework for solving the Maximum Clique Problem by reformulating it as a vertex ordering task. Instead of classifying nodes or using heuristic degree-based orderings, the model learns an optimal permutation of vertices that brings clique nodes to the front of the adjacency matrix. A graph neural network generates a soft permutation matrix trained via the Gumbel–Sinkhorn operator to approximate a valid ordering, while a Chebyshev distance–based loss encourages clique vertices to cluster in the top-left corner. The learned ordering is then integrated into the classical MaxCliqueDyn algorithm, leading to faster search convergence and reduced computational cost without requiring labeled data.

**Strengths:**

1. The author presents a novel reformulation of the Maximum Clique Problem as an unsupervised vertex-ordering task, combining permutation learning with classical optimization in an elegant and creative way.

2. Theoretical reasoning and experiments are solid, with clear links between the objective function and clique geometry, and measurable gains in solver efficiency.
Writing is clear and well structured, with intuitive explanations and effective visualizations that make complex ideas accessible.

3. This paper demonstrates that unsupervised geometric learning can improve the efficiency of deterministic solvers, suggesting a generalizable paradigm applicable to other NP-hard problems such as graph coloring or independent set detection.

**Weaknesses:**

1. The evaluation is restricted to synthetic graphs, which, while controlled, do not fully represent real-world graph structures such as social, biological, or citation networks. Including results on more heterogeneous datasets would strengthen claims of generalization and robustness.

2. Although the method performs well on graphs up to 200 nodes, the paper does not explore computational limits for larger graphs, where permutation learning and Sinkhorn iterations may become expensive. An analysis of complexity or scalability curves would add practical depth.

3. The paper could benefit from more detailed ablation experiments to isolate the contribution of each design choice—such as the specific form of the Chebyshev-based weighting matrix or the impact of Gumbel noise magnitude on learning stability.

**Questions:**

1. Have the authors tested the model on structured real-world graphs (e.g., citation, protein–protein interaction, or social networks)? Since Erdős–Rényi graphs lack community structure, results on more realistic data could clarify how well the learned ordering adapts to heterogeneous topologies.

2. How does the computational cost of the Gumbel–Sinkhorn iterations scale with graph size? For larger graphs (e.g., >500 nodes), does the model remain efficient, or does the continuous permutation approximation become a bottleneck?

3. Is the proposed ordering applicable to other exact algorithms for the Maximum Clique Problem or to related NP-hard problems (like graph coloring or independent set)? Some experimental or conceptual discussion would help clarify its general utility.

---

> ### Author Response · Authors · 2025-11-19
> **Response to Reviewer TboW**
>
> Dear Reviewer TboW, thank you for your comments. Below is our detailed response to your questions.
>
> -----
>
> *Weakness 1 and Question 1: Have the authors tested the model on structured real-world graphs (e.g., citation, protein–protein interaction, or social networks)? Since Erdős–Rényi graphs lack community structure, results on more realistic data could clarify how well the learned ordering adapts to heterogeneous topologies.*
>
> Answer: here we evaluated our method on additional graph models beyond the Erdős–Rényi (ER) family. Specifically, we tested on:
>
> - Barabási–Albert (BA) graphs with parameters ( n = 200 , m = 20 )
> - Watts–Strogatz (WS) graphs with parameters ( n = 20, k = 20 , p = 1 )
>
> On these non-ER graph types, our learned clique-oriented ordering continued to provide improvements over traditional degree-based ordering within the MaxCliqueDyn framework. We observed:
>
> - 1.05% reduction in computation time on Barabási–Albert graphs
> - 2.98% reduction in computation time on Watts–Strogatz graphs
>
> These results demonstrate that our method generalizes well across different graph topologies and maintains efficiency improvements beyond ER graphs.
>
> We further test our method on several instances taken from the DIMACS benchmark set. The selected instances include:
>
> - brock200_4.clq
> - brock200_3.clq
> - brock200_2.clq
> - brock200_1.clq
> - c-fat200-5.clq
> - c-fat200-2.clq
> - c-fat200-1.clq
> - san200_0.9_3.clq
> - san200_0.9_2.clq
> - san200_0.9_1.clq
> - san200_0.7_2.clq
> - san200_0.7_1.clq
> - sanr200_0.9.clq
> - sanr200_0.7.clq
>
> ### Baseline Results
>
> - Average steps: 498,405
> - Average time: 1.306 seconds
>
> ### Results Using Our UL Method
>
> After training our UL-based model on instances with similar density, we apply it to the same DIMACS benchmark instances. The results are:
>
> - Average steps: 489,350
> - Average time: 1.265 seconds
>
> These results suggest that our model generalizes well to unseen, real-world instances.
>
> -----
> *Weakness 2 and Question 2: …computational limits for larger graphs, where permutation learning and Sinkhorn iterations may become expensive. An analysis of complexity or scalability curves would add practical depth….How does the computational cost of the Gumbel–Sinkhorn iterations scale with graph size? For larger graphs (e.g., >500 nodes), does the model remain efficient, or does the continuous permutation approximation become a bottleneck?*
>
> Answer: the continuous permutation approximation is not a bottleneck, in our code, we are using Pytorch, we can use torch.complie to accelerate it, so the code is like:
>
> ```python
> def fast_sinkhorn_iteration(Z, n_iter=10):
>     Z = Z.to(torch.float32)
>     for _ in range(n_iter):
>         Z = Z - torch.logsumexp(Z, dim=-1, keepdim=True)
>         Z = Z - torch.logsumexp(Z, dim=-2, keepdim=True)
>     return Z
>
> # Torch 2.x: compile the function
> compiled_fast_sinkhorn = torch.compile(fast_sinkhorn_iteration)
> ```
>
> We evaluate the speed using 100 iterations on a problem size of 500. We repeat this process 20 times and report the mean and standard deviation. The results are as follows:
>
> Mean: 4.01ms; std:0.40ms; min value: 2.70ms, max value: 4.15ms.
>
> For the permutation step, we use the torch-linear-assignment package to obtain a hard permutation from the soft permutation. Specifically, we run the following procedure and report the average over 20 runs.
>
> ```python
> import torch
> from torch_linear_assignment import batch_linear_assignment
>
> # cost matrix, batch size = 256
> cost = torch.rand(256, 500, 500)
>
> start = time.perf_counter()
> _ = batch_linear_assignment(cost)
> end = time.perf_counter()
> ```
>
> The results are as follows:
> mean: 7.80 ms
> std:  0.048 ms
> min:  7.69 ms
> max:  7.88 ms
>
> **Overall, we can see that the costs of the sinkhorn and soft to hard permutation steps are marginal compared with the solver’s running time.**

---

> > ### Author Response · Authors · 2025-11-19
> > **Response to Reviewer TboW**
> >
> > *Weakness 3: The paper could benefit from more detailed ablation experiments to isolate the contribution of each design choice—such as the specific form of the Chebyshev-based weighting matrix or the impact of Gumbel noise magnitude on learning stability.*
> >
> > Answer: In our ablation study, we find that the Gumbel noise has only a minor effect on the results. For example, for the setting with $n = 200$ and $p = 0.9$, using a Sinkhorn temperature of $4.0$ with noise levels of $0.01$ and $0.05$ both lead to roughly a 2.5 \% reduction in total runtime. The most influential parameter is the weighting in the Chebyshev-based matrix $\mathcal{D}_{\mathrm{clique}}$.
> >
> > In our paper, we set
> >
> > $$
> > \mathcal{D}_{\mathrm{clique}} = (1 + \epsilon)^{(C_n - n/2)},
> > $$
> >
> > where $\epsilon$ is a positive constant. In our experiments, $\epsilon$ is chosen to be relatively small. When we tried a larger value, for example, $\epsilon = 0.5$ on graphs of size $100$, the model became unstable during training and sometimes produced results worse than simple degree-based sorting. We believe this is due to numerical issues caused by the rapid exponential growth in $\mathcal{D}_{\mathrm{clique}}$.
> >
> > This problem can be mitigated by using a smaller $\epsilon$, or by redefining $\mathcal{D}_{\mathrm{clique}}$ to grow more slowly, possibly using a polynomial instead of an exponential form, as long as the weighting still emphasizes the top-left region to encourage clique formation. For instance, one could consider something like $(\bar{C}_n - n/2)^2$ or another polynomial weighting.
> >
> > In practice, our current choice works well, and we believe that any formulation that increases weights toward the top left while avoiding numerical explosion should be sufficient.
> >
> >
> > -----
> > *Question 3: Is the proposed ordering applicable to other exact algorithms for the Maximum Clique Problem or to related NP-hard problems (like graph coloring or independent set)? Some experimental or conceptual discussion would help clarify its general utility.*
> >
> > Answer: Most exact MCP solvers, e.g., Tomita & Seki (MCQ, MCT, MCR), BBMC, MaxCliqueDyn, branch-and-cut variants, and coloring-based pruning algorithms, all share a similiar architecture, and they are heavily on an initial vertex ordering. For example, in the “A new exact maximum clique algorithm for large and massive sparse graphs” by Segundo et al. the author use minimum-degree-last ordering.
> >
> > Our model proposes a learned clique-oriented ordering that replaces only the initial ordering stage in MaxCliqueDyn. Because this initial ordering step is conceptually the same across most exact MCP solvers, the method should plug into many of them:
> >
> > Our method can be adapted to:
> >
> > - **Tomita’s MCQ/MCS/MCR algorithms**
> >
> >     These rely on ordering of vertices before color bounding. Replacing degree-sort with a learned ordering is straightforward.
> >
> > - **BBMC and BBMC variants**
> >
> >     Again, these depend on a static initial order before bitset operations.
> >
> > - **Konc & Janežič’s MaxCliqueDyn+EFL+SCR**
> >
> >     Since this extends MaxCliqueDyn, the same reasoning applies.
> >
> > - **Branch-and-cut frameworks**
> >
> >     The branching order (node selection in the search tree) can be supplied by the learned ordering.
> >
> >
> > As future work, our learned permutation-based ordering could be integrated with other exact Maximum Clique frameworks that currently lack dedicated ordering heuristics, such as the MaxSAT-INC algorithm[1]—which strengthens clique search through MaxSAT-derived incremental upper bounds—and the ICTAI’10 algorithm[2], which is a coloring-based branch-and-bound method that relies on recursive expansion and SAT-inspired pruning rules. In both cases, our ordering can be added as a simple preprocessing step that replaces the solver’s initial vertex order before their respective pruning or coloring mechanisms are applied. We expect that introducing our learned ordering would enable stronger early lower bounds, improve the effectiveness of their upper-bound computations, and reduce the overall search tree size without requiring any modifications to their internal logic.
> >
> > [1] Li, Chu-Min, Zhiwen Fang, and Ke Xu. "Combining MaxSAT reasoning and incremental upper bound for the maximum clique problem." 2013 IEEE 25th international conference on tools with artificial intelligence. IEEE, 2013.
> >
> > [2] Li, Chu-Min, and Zhe Quan. "Combining graph structure exploitation and propositional reasoning for the maximum clique problem." 2010 22nd IEEE international conference on tools with artificial intelligence. Vol. 1. IEEE, 2010.

---

> > > ### Comment · Reviewer_TboW · 2025-11-27
> > >
> > > Thank you for the detailed response, I tend to keep my positive score.

---

### Meta-Review · Area_Chair_YufB · 2025-12-28

**Summary:**

The main concerns are about the experiment design, especially the choice of the baseline and the dataset. I think these concerns are major, because the main result this work can only be evaluated via experiments instead of theoretical arguments. The authors made an attempt to address the issue in the rebuttal, but they do not seem to be convincing enough to me. This leads to reject.

**Reviewer Concerns:**

I see the following major issues raised by the reviewers, mostly on experiments.

1. The small dataset size and limited dataset especially no real-world dataset evaluation.
2. More baseline and downstream algorithms should be added into experiments.

For 1, the authors added more synthesized datasets, as well as a real dataset DIMACS. I think this helps, but is not comprehensive enough. Furthermore, the results (improvement of running time) on DIMACS does not seem to be very competitive.

For 2, while authors promised to add more evaluation, they nonetheless did not provide convincing evidence (e.g., some preliminary experiment results) on how the result goes.

Overall, both issues are not adequately addressed.

**Reviewer Scores:**

I think all reviewers would keep their score, because the rebuttal does not provide significant materials.

---

### Decision · Program_Chairs · 2026-01-26

Reject